# Better Exploration with Optimistic Actor-Critic

**Kamil Ciosek**
Microsoft Research Cambridge, UK
`kamil.ciosek@microsoft.com`

**Quan Vuong**[*]
University of California San Diego
`qvuong@ucsd.edu`

**Robert Loftin**
Microsoft Research Cambridge, UK
`t-roloft@microsoft.com`

**Katja Hofmann**
Microsoft Research Cambridge, UK
`katja.hofmann@microsoft.com`

## Abstract

Actor-critic methods, a type of model-free Reinforcement Learning, have been successfully applied to challenging tasks in continuous control, often achieving state-of-the art performance. However, wide-scale adoption of these methods in real-world domains is made difficult by their poor sample efficiency. We address this problem both theoretically and empirically. On the theoretical side, we identify two phenomena preventing efficient exploration in existing state-of-the-art algorithms such as Soft Actor Critic. First, combining a greedy actor update with a pessimistic estimate of the critic leads to the avoidance of actions that the agent does not know about, a phenomenon we call *pessimistic underexploration*. Second, current algorithms are *directionally uninformed*, sampling actions with equal probability in opposite directions from the current mean. This is wasteful, since we typically need actions taken along certain directions much more than others. To address both of these phenomena, we introduce a new algorithm, Optimistic Actor Critic, which approximates a lower and upper confidence bound on the state-action value function. This allows us to apply the principle of *optimism in the face of uncertainty* to perform directed exploration using the upper bound while still using the lower bound to avoid overestimation. We evaluate OAC in several challenging continuous control tasks, achieving state-of the art sample efficiency.

## 1 Introduction

A major obstacle that impedes a wider adoption of actor-critic methods [31, 40, 49, 44] for control tasks is their poor sample efficiency. In practice, despite impressive recent advances [24, 17], millions of environment interactions are needed to obtain a reasonably performant policy for control problems with moderate complexity. In systems where obtaining samples is expensive, this often makes the deployment of these algorithms prohibitively costly.

This paper aims at mitigating this problem by more efficient exploration . We begin by examining the exploration behavior of SAC [24] and TD3 [17], two recent model-free algorithms with state-of-the-art sample efficiency and make two insights. First, in order to avoid overestimation [26, 46], SAC and TD3 use a critic that computes an approximate lower confidence bound[2]. The actor then adjusts the exploration policy to maximize this lower bound. This improves the stability of the updates and allows the use of larger learning rates. However, using the lower bound can also seriously inhibit exploration if it is far from the true Q-function. If the lower bound has a spurious maximum, the covariance of the policy will decrease, causing *pessimistic underexploration*, i.e. discouraging

---

[*]Work done while an intern at Microsoft Research, Cambridge.
[2]See Appendix C for details.

the algorithm from sampling actions that would lead to an improvement to the flawed estimate of the critic. Moreover, Gaussian policies are *directionally uninformed*, sampling actions with equal probability in any two opposing directions from the mean. This is wasteful since some regions in the action space close to the current policy are likely to have already been explored by past policies and do not require more samples.

We formulate Optimistic Actor-Critic (OAC), an algorithm which explores more efficiently by applying the principle of optimism in the face of uncertainty [9]. OAC uses an off-policy exploration strategy that is adjusted to maximize an upper confidence bound to the critic, obtained from an epistemic uncertainty estimate on the Q-function computed with the bootstrap [35]. OAC avoids pessimistic underexploration because it uses an upper bound to determine exploration covariance. Because the exploration policy is not constrained to have the same mean as the target policy, OAC is directionally informed, reducing the waste arising from sampling parts of action space that have already been explored by past policies.

Off-policy Reinforcement Leaning is known to be prone to instability when combined with function approximation, a phenomenon known as the *deadly triad* [43, 47]. OAC achieves stability by enforcing a KL constraint between the exploration policy and the target policy. Moreover, similarly to SAC and TD3, OAC mitigates overestimation by updating its target policy using a lower confidence bound of the critic [26, 46].

Empirically, we evaluate Optimistic Actor Critic in several challenging continuous control tasks and achieve state-of-the-art sample efficiency on the Humanoid benchmark. We perform ablations and isolate the effect of bootstrapped uncertainty estimates on performance. Moreover, we perform hyperparameter ablations and demonstrate that OAC is stable in practice.

## 2  Preliminaries

*Reinforcement learning* (RL) aims to learn optimal behavior policies for an agent acting in an environment with a scalar reward signal. Formally, we consider a Markov decision process [39], defined as a tuple $(S, A, R, p, p_0, \gamma)$. An agent observes an environmental state $s \in S = \mathbb{R}^n$; takes a sequence of actions $a_1, a_2, ...$, where $a_t \in A \subseteq \mathbb{R}^d$; transitions to the next state $s' \sim p(\cdot|s, a)$ under the state transition distribution $p(s'|s, a)$; and receives a scalar reward $r \in \mathbb{R}$. The agent's initial state $s_0$ is distributed as $s_0 \sim p_0(\cdot)$.

A policy $\pi$ can be used to generate actions $a \sim \pi(\cdot|s)$. Using the policy to sequentially generate actions allows us to obtain a trajectory through the environment $\tau = (s_0, a_0, r_0, s_1, a_1, r_1, ...)$. For any given policy, we define the action-value function as $Q^\pi(s, a) = E_{\tau:s_0=s,a_0=a}[\sum_t \gamma^t r_t]$, where $\gamma \in [0, 1)$ is a discount factor. We assume that $Q^\pi(s, a)$ is differentiable with respect to the action. The objective of Reinforcement Learning is to find a deployment policy $\pi_{\text{eval}}$ which maximizes the total return $J = E_{\tau:s_0 \sim p_0}[\sum_t \gamma^t r_t]$. In order to provide regularization and aid exploration, most actor-critic algorithms [24, 17, 31] do not adjust $\pi_{\text{eval}}$ directly. Instead, they use a target policy $\pi_T$, trained to have high entropy in addition to maximizing the expected return $J$.[3] The deployment policy $\pi_{\text{eval}}$ is typically deterministic and set to the mean of the stochastic target policy $\pi_T$.

Actor-critic methods [44, 6, 8, 7] seek a locally optimal target policy $\pi_T$ by maintaining a *critic*, learned using a value-based method, and an *actor*, adjusted using a policy gradient update. The critic is learned with a variant of SARSA [48, 43, 41]. In order to limit overestimation [26, 46], modern actor-critic methods learn an approximate lower confidence bound on the Q-function [24, 17], obtained by using two networks $\hat{Q}_{\text{LB}}^1$ and $\hat{Q}_{\text{LB}}^2$, which have identical structure, but are initialized with different weights. In order to avoid cumbersome terminology, we refer to $\hat{Q}_{\text{LB}}$ simply as a lower bound in the remainder of the paper. Another set of target networks [33, 31] slowly tracks the values of $\hat{Q}_{\text{LB}}$ in order to improve stability.

$$\hat{Q}_{\text{LB}}(s_t, a_t) = \min(\hat{Q}_{\text{LB}}^1(s_t, a_t), \hat{Q}_{\text{LB}}^2(s_t, a_t)) \tag{1}$$

$$\hat{Q}_{\text{LB}}^{\{1,2\}}(s_t, a_t) \leftarrow R(s_t, a_t) + \gamma \min(\check{Q}_{\text{LB}}^1(s_{t+1}, a), \check{Q}_{\text{LB}}^2(s_{t+1}, a)) \text{ where } a \sim \pi_T(\cdot|s_{t+1}). \tag{2}$$

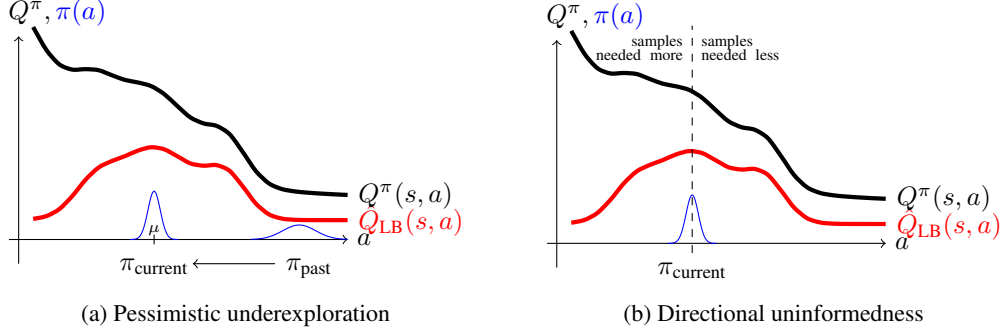

(a) Pessimistic underexploration       (b) Directional uninformedness

Figure 1: Exploration inefficiencies in actor-critic methods. The state $s$ is fixed. The graph shows $Q^\pi$, which is unknown to the algorithm, its known lower bound $\hat{Q}_{\text{LB}}$ (in red) and two policies $\pi_{\text{current}}$ and $\pi_{\text{past}}$ at different time-steps of the algorithm (in blue).

Meanwhile, the actor adjusts the policy parameter vector $\theta$ of the policy $\pi_T$ in order to maximize $J$ by following its gradient. The gradient can be written in several forms [44, 40, 13, 27, 22, 23]. Recent actor-critic methods use a reparametrised policy gradient [27, 22, 23]. We denote a random variable sampled from a standard multivariate Gaussian as $\varepsilon \sim \mathcal{N}(0, I)$ and denote the standard normal density as $\phi(\varepsilon)$. The re-parametrisation function $f$ is defined such that the probability density of the random variable $f_\theta(s, \varepsilon)$ is the same as the density of $\pi_T(a|s)$, where $\varepsilon \sim \mathcal{N}(0, I)$. The gradient of the return can then be written as:

$$\nabla_\theta J = \int_s \rho(s) \int_\varepsilon \nabla_\theta \hat{Q}_{\text{LB}}(s, f_\theta(s, \varepsilon))\phi(\varepsilon)d\varepsilon ds \tag{3}$$

where $\rho(s) \triangleq \sum_{t=0}^\infty \gamma^t p(s_t = s|s_0)$ is the discounted-ergodic occupancy measure. In order to provide regularization and encourage exploration, it is common to use a gradient $\nabla_\theta J^\alpha$ that adds an additional entropy term $\nabla_\theta \mathcal{H}(\pi(\cdot, s))$.

$$\nabla_\theta J^\alpha_{\hat{Q}_{\text{LB}}} = \int_s \rho(s) \int_\varepsilon \nabla_\theta \hat{Q}_{\text{LB}}(s, f_\theta(s, \varepsilon))\phi(\varepsilon)d\varepsilon + \alpha \underbrace{\int_\varepsilon -\nabla_\theta \log f_\theta(s, \varepsilon)\phi(\varepsilon)d\varepsilon}_{\nabla_\theta \mathcal{H}(\pi(\cdot, s))} ds \tag{4}$$

During training, (4) is approximated with samples by replacing integration over $\varepsilon$ with Monte-Carlo estimates and integration over the state space with a sum along the trajectory.

$$\nabla_\theta J^\alpha_{\hat{Q}_{\text{LB}}} \approx \nabla_\theta \hat{J}^\alpha_{\hat{Q}_{\text{LB}}} = \sum_{t=0}^N \gamma^t \nabla_\theta \hat{Q}_{\text{LB}}(s_t, f_\theta(s, \varepsilon_t)) + \alpha - \nabla_\theta \log f_\theta(s_t, \varepsilon_t). \tag{5}$$

In the standard set-up, actions used in (1) and (5) are generated using $\pi_T$. In the table-lookup case, the update can be reliably applied off-policy, using an action generated with a separate exploration policy $\pi_E$. In the function approximation setting, this leads to updates that can be biased because of the changes to $\rho(s)$. In this work, we address these issues by imposing a KL constraint between the exploration policy and the target policy. We give a more detailed account of addressing the associated stability issues in section 4.3.

## 3 Existing Exploration Strategy is Inefficient

As mentioned earlier, modern actor-critic methods such as SAC [24] and TD3 [17] explore in an inefficient way. We now give more details about the phenomena that lead to this inefficiency.

**Pessimistic underexploration.** In order to improve sample efficiency by preventing the catastrophic overestimation of the critic [26, 46], SAC and TD3 [17, 25, 24] use a lower bound approximation to the critic, similar to (1). However, relying on this lower bound for exploration is inefficient. By greedily maximizing the lower bound, the policy becomes very concentrated near a maximum. When the critic is inaccurate and the maximum is spurious, this can be very harmful. This is illustrated in Figure 1a. At first, the agent explores with a broad policy, denoted $\pi_{\text{past}}$. Since $\hat{Q}_{\text{LB}}$ increases to the left, the policy gradually moves in that direction, becoming $\pi_{\text{current}}$. Because $\hat{Q}_{\text{LB}}$ (shown in red)

has a maximum at the mean $\mu$ of $\pi_{\text{current}}$, the policy $\pi_{\text{current}}$ has a small standard deviation. This is suboptimal since we need to sample actions far away from the mean to find out that the true critic $Q^\pi$ does not have a maximum at $\mu$. We include evidence that this problem actually happens in MuJoCo Ant in Appendix F.

The phenomenon of underexploration is specific to the lower as opposed to an upper bound. An upper bound which is too large in certain areas of the action space encourages the agent to explore them and correct the critic, akin to optimistic initialization in the tabular setting [42, 43]. We include more intuition about the difference between the upper and lower bound in Appendix I. Due to overestimation, we cannot address pessimistic underexploration by simply using the upper bound in the actor [17]. Instead, recent algorithms have used an entropy term (4) in the actor update. While this helps exploration somewhat by preventing the covariance from collapsing to zero, it does not address the core issue that we need to explore more around a spurious maximum. We propose a more effective solution in section 4.

**Directional uninformedness.** Actor-critic algorithms that use Gaussian policies, like SAC [25] and TD3 [17], sample actions in opposite directions from the mean with equal probability. However, in a policy gradient algorithm, the current policy will have been obtained by incremental updates, which means that it won't be very different from recent past policies. Therefore, exploration in both directions is wasteful, since the parts of the action space where past policies had high density are likely to have already been explored. This phenomenon is shown in Figure 1b. Since the policy $\pi_{\text{current}}$ is Gaussian and symmetric around the mean, it is equally likely to sample actions to the left and to the right. However, while sampling to the left would be useful for learning an improved critic, sampling to the right is wasteful, since the critic estimate in that part of the action space is already good enough. In section 4, we address this issue by using an exploration policy shifted relative to the target policy.

# 4 Better Exploration with Optimism

Optimistic Actor Critic (OAC) is based on the principle of optimism in the face of uncertainty [50]. Inspired by recent theoretical results about efficient exploration in model-free RL [28], OAC obtains an exploration policy $\pi_E$ which locally maximizes an approximate upper confidence bound of $Q^\pi$ each time the agent enters a new state. The policy $\pi_E$ is separate from the target policy $\pi_T$ learned using (5) and is used only to sample actions in the environment. Formally, the exploration policy $\pi_E = \mathcal{N}(\mu_E, \Sigma_E)$, is defined as

$$\mu_e, \Sigma_E = \underset{\substack{\mu, \Sigma: \\ \text{KL}(\mathcal{N}(\mu,\Sigma), \mathcal{N}(\mu_T, \Sigma_T)) \leq \delta}}{\arg\max} E_{a \sim \mathcal{N}(\mu,\Sigma)} \left[ \bar{Q}_{\text{UB}}(s, a) \right]. \tag{6}$$

Below, we derive the OAC algorithm formally. We begin by obtaining the upper bound $\bar{Q}_{\text{UB}}(s, a)$ (section 4.1). We then motivate the optimization problem (6), in particular the use of the KL constraint (section 4.2). Finally, in section 4.3, we describe the OAC algorithm and outline how it mitigates *pessimistic underexploration* and *directional uninformedness* while still maintaining the stability of learning. In Section 4.4, we compare OAC to related work. In Appendix B, we derive an alternative variant of OAC that works with deterministic policies.

## 4.1 Obtaining an Upper Bound

The approximate upper confidence bound $\bar{Q}_{\text{UB}}$ used by OAC is derived in three stages. First, we obtain an epistemic uncertainty estimate $\sigma_Q$ about the true state-action value function $Q$. We then use it to define an upper bound $\hat{Q}_{\text{UB}}$. Finally, we introduce its linear approximation $\bar{Q}_{\text{UB}}$, which allows us to obtain a tractable algorithm.

**Epistemic uncertainty** For computational efficiency, we use a Gaussian distribution to model epistemic uncertainty. We fit mean and standard deviation based on bootstraps [16] of the critic. The mean belief is defined as $\mu_Q(s, a) = \frac{1}{2}\left(\hat{Q}_{\text{LB}}^1(s, a) + \hat{Q}_{\text{LB}}^2(s, a)\right)$, while the standard deviation is

$$\sigma_Q(s, a) = \sqrt{\sum_{i \in \{1,2\}} \frac{1}{2}\left(\hat{Q}_{\text{LB}}^i(s, a) - \mu_Q(s, a)\right)^2} = \frac{1}{2}\left|\hat{Q}_{\text{LB}}^1(s, a) - \hat{Q}_{\text{LB}}^2(s, a)\right|. \tag{7}$$

Here, the second equality is derived in appendix C. The bootstraps are obtained using (1). Since existing algorithms [24, 17] already maintain two bootstraps, we can obtain $\mu_Q$ and $\sigma_Q$ at negligible computational cost. Despite the fact that (1) uses the same target value for both bootstraps, we demonstrate in Section 5 that using a two-network bootstrap leads to a large performance improvement in practice. Moreover, OAC can be easily extended to to use more expensive and better uncertainty estimates if required.

**Upper bound.** Using the uncertainty estimate (24), we define the upper bound as $\hat{Q}_{\text{UB}}(s,a) = \mu_Q(s,a) + \beta_{\text{UB}}\sigma_Q(s,a)$. We use the parameter $\beta_{\text{UB}} \in \mathbb{R}^+$ to fix the level of optimism. In order to obtain a tractable algorithm, we approximate $\hat{Q}_{\text{UB}}$ with a linear function $\bar{Q}_{\text{UB}}$.

$$\bar{Q}_{\text{UB}}(s,a) = a^\top \left[\nabla_a \hat{Q}_{\text{UB}}(s,a)\right]_{a=\mu_T} + \text{const} \tag{8}$$

By Taylor's theorem, $\bar{Q}_{\text{UB}}(s,a)$ is the best possible linear fit to $\hat{Q}_{\text{UB}}(s,a)$ in a sufficiently small region near the current policy mean $\mu_T$ for any fixed state $s$ [10, Theorem 3.22]. Since the gradient $\left[\nabla_a \hat{Q}_{\text{UB}}(s,a)\right]_{a=\mu_T}$ is computationally similar to the lower-bound gradients in (5), our upper bound estimate can be easily obtained in practice without additional tuning.

## 4.2 Optimistic Exploration

Our exploration policy $\pi_E$, introduced in (6), trades off between two criteria: the maximization of an upper bound $\bar{Q}_{\text{UB}}(s,a)$, defined in (8), which increases our chances of executing informative actions, according to the principle of *optimism in the face of uncertainty* [9], and constraining the maximum KL divergence between the exploration policy and the target policy $\pi_T$, which ensures the stability of updates. The KL constraint in (6) is crucial for two reasons. First, it guarantees that the exploration policy $\pi_E$ is not very different from the target policy $\pi_T$. This allows us to preserve the stability of optimization and makes it less likely that we take catastrophically bad actions, ending the episode and preventing further learning. Second, it makes sure that the exploration policy remains within the action range where the approximate upper bound $\bar{Q}_{\text{UB}}$ is accurate. We chose the KL divergence over other similarity measures for probability distributions since it leads to tractable updates.

Thanks to the linear form on $\bar{Q}_{\text{UB}}$ and because both $\pi_E$ and $\pi_T$ are Gaussian, the maximization of (6) can be solved in closed form. We state the solution below.

**Proposition 1.** *The exploration policy resulting from* (6) *has the form* $\pi_E = \mathcal{N}(\mu_E, \Sigma_E)$, *where*

$$\mu_E = \mu_T + \frac{\sqrt{2\delta}}{\left\|\left[\nabla_a \hat{Q}_{UB}(s,a)\right]_{a=\mu_T}\right\|_\Sigma} \Sigma_T \left[\nabla_a \hat{Q}_{UB}(s,a)\right]_{a=\mu_T} \quad and \quad \Sigma_E = \Sigma_T. \tag{9}$$

We stress that the covariance of the exploration policy is the same as the target policy. The proof is deferred to Appendix A.

## 4.3 The Optimistic Actor-Critic Algorithm

Optimistic Actor Critic (see Algorithm 1) samples actions using the exploration policy (9) in line 4 and stores it in a memory buffer. The term $\left[\nabla_a \hat{Q}_{\text{UB}}(s,a)\right]_{a=\mu_T}$ in (9) is computed at minimal cost[4] using automatic differentiation, analogous to the critic derivative in the actor update (4). OAC then uses its memory buffer to train the critic (line 10) and the actor (line 12). We also introduced a modification of the lower bound used in the actor, using $\hat{Q}'_{\text{LB}} = \mu_Q(s,a) + \beta_{\text{LB}}\sigma_Q(s,a)$, allowing us to use more conservative policy updates. The critic (1) is recovered by setting $\beta_{\text{LB}} = -1$.

**OAC avoids the pitfalls of greedy exploration** Figure 2 illustrates OAC's exploration policy $\pi_E$ visually. Since the policy $\pi_E$ is far from the spurious maximum of $\hat{Q}_{\text{LB}}$ (red line in figure 2), executing actions sampled from $\pi_E$ leads to a quick correction to the critic estimate. This way, *OAC avoids pessimistic underexploration*. Since $\pi_E$ is not symmetric with respect to the mean of $\pi_T$ (dashed line), *OAC also avoids directional uninformedness*.

**Algorithm 1** Optimistic Actor-Critic (OAC).

---

**Require:** $w_1, w_2, \theta$          ▷ Initial parameters $w_1, w_2$ of the critic and $\theta$ of the target policy $\pi_T$.
1:   $\breve{w}_1 \leftarrow w_1, \breve{w}_2 \leftarrow w_2, \mathcal{D} \leftarrow \emptyset$        ▷ Initialize target network weights and replay pool
2:   **for** each iteration **do**
3:      **for** each environment step **do**
4:         $a_t \sim \pi_E(a_t|s_t)$            ▷ Sample action from exploration policy as in (9).
5:         $s_{t+1} \sim p(s_{t+1}|s_t, a_t)$        ▷ Sample transition from the environment
6:         $\mathcal{D} \leftarrow \mathcal{D} \cup \{(s_t, a_t, R(s_t, a_t), s_{t+1})\}$     ▷ Store the transition in the replay pool
7:      **end for**
8:      **for** each training step **do**
9:         **for** $i \in \{1, 2\}$ **do**             ▷ Update two bootstraps of the critic
10:            update $w_i$ with $\hat{\nabla}_{w_i} \| \hat{Q}^i_{\text{LB}}(s_t, a_t) - R(s_t, a_t) - \gamma \min(\breve{Q}^1_{\text{LB}}(s_{t+1}, a), \breve{Q}^2_{\text{LB}}(s_{t+1}, a)) \|^2_2$
11:         **end for**
12:         update $\theta$ with $\nabla_\theta \hat{J}^\alpha_{\hat{Q}'_{\text{LB}}}$         ▷ Policy gradient update.
13:         $\breve{w}_1 \leftarrow \tau w_1 + (1 - \tau)\breve{w}_1, \breve{w}_2 \leftarrow \tau w_2 + (1 - \tau)\breve{w}_2$     ▷ Update target networks
14:      **end for**
15: **end for**
**Output:** $w_1, w_2, \theta$                           ▷ Optimized parameters

---

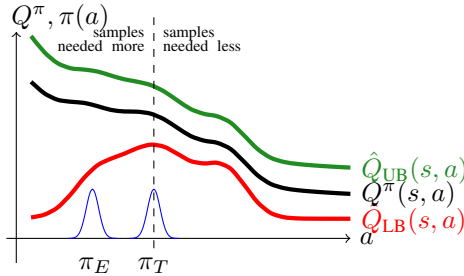

Figure 2: The OAC exploration policy $\pi_E$ avoids pessimistic underexploration by sampling far from the spurious maximum of the lower bound $\hat{Q}_{\text{LB}}$. Since $\pi_E$ is not symmetric wrt. the mean of the target policy (dashed line), it also addresses directional uninformedness.

**Stability**     While off-policy deep Reinforcement Learning is difficult to stabilize in general [43, 47], OAC is remarkably stable. Due to the KL constraint in equation (6), the exploration policy $\pi_E$ remains close to the target policy $\pi_T$. In fact, despite using a separate exploration policy, OAC isn't very different in this respect from SAC [24] or TD3 [17], which explore with a stochastic policy but use a deterministic policy for evaluation. In Section 5, we demonstrate empirically that OAC and SAC are equally stable in practice. Moreover, similarly to other recent state-of-the-art actor-critic algorithms [17, 25], we use target networks [33, 31] to stabilize learning. We provide the details in Appendix D.

**Overestimation vs Optimism**     While OAC is an optimistic algorithm, it does not exhibit catastrophic overestimation [17, 26, 46]. OAC uses the optimistic estimate (8) for exploration only. The policy $\pi_E$ is computed from scratch (line 4 in Algorithm 1) every time the algorithm takes an action and is used only for exploration. The critic and actor updates (1) and (5) are still performed with a lower bound. This means that there is no way the upper bound can influence the critic except indirectly through the distribution of state-action pairs in the memory buffer.

## 4.4   Related work

OAC is distinct from other methods that maintain uncertainty estimates over the state-action value function. Actor-Expert [32] uses a point estimate of $Q^\star$, unlike OAC, which uses a bootstrap approximating $Q^\pi$. Bayesian actor-critic methods [19–21] model the probability distribution over $Q^\pi$, but unlike OAC, do not use it for exploration. Approaches combining DQN with bootstrap [11, 36] and the uncertainty Bellman equation [34] are designed for discrete actions. Model-based reinforcement

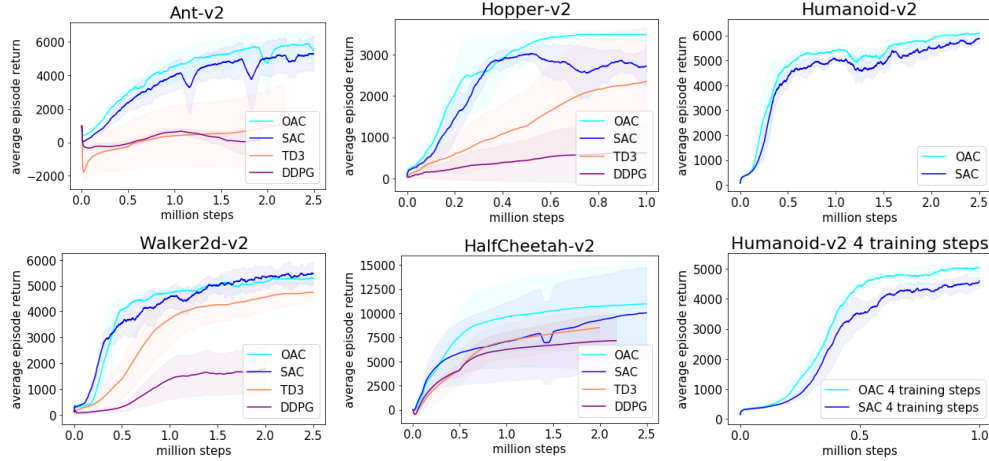

Figure 3: OAC versus SAC, TD3, DDPG on 5 Mujoco environments. The horizontal axis indicates number of environment steps. The vertical axis indicates the total undiscounted return. The shaded areas denote one standard deviation.

learning methods thet involve uncertainty [18, 15, 12] are very computationally expensive due to the need of learning a distribution over environment models. OAC may seem superficially similar to natural actor critic [5, 29, 37, 38] due to the KL constraint in (6). In fact, it is very different. While natural actor critic uses KL to enforce the similarity between infinitesimally small updates to the target policy, OAC constrains the *exploration policy* to be within a non-trivial distance of the target policy. Other approaches that define the exploration policy as a solution to a KL-constrained optimization problem include MOTO [2], MORE [4] and Maximum a Posteriori Policy optimization [3]. These methods differ from OAC in that they do not use epistemic uncertainty estimates and explore by enforcing entropy.

## 5   Experiments

Our experiments have three main goals. First, to test whether Optimistic Actor Critic has performance competitive to state-of-the art algorithms. Second, to assess whether optimistic exploration based on the bootstrapped uncertainty estimate (24), is sufficient to produce a performance improvement. Third, to assess whether optimistic exploration adversely affects the stability of the learning process.

**MuJoCo Continuous Control**   We test OAC on the MuJoCo [45] continuous control benchmarks. We compare OAC to SAC [25] and TD3 [17], two recent model-free RL methods that achieve state-of-the art performance. For completeness, we also include a comparison to a tuned version of DDPG [31], an established algorithm that does not maintain multiple bootstraps of the critic network. OAC uses 3 hyper-parameters related to exploration. The parameters $\beta_{\mathrm{UB}}$ and $\beta_{\mathrm{LB}}$ control the amount of uncertainty used to compute the upper and lower bound respectively. The parameter $\delta$ controls the maximal allowed divergence between the exploration policy and the target policy. We provide the values of all hyper-parameters and details of the hyper-parameter tuning in Appendix D. Results in Figure 3 show that using optimism improves the overall performance of actor-critic methods. On Ant, OAC improves the performance somewhat. On Hopper, OAC achieves state-of the art final performance. On Walker, we achieve the same performance as SAC while the high variance of results on HalfCheetah makes it difficult to draw conclusions on which algorithm performs better.[5].

**State-of-the art result on Humanoid**   The upper-right plot of Figure 3 shows that the vanilla version of OAC outperforms SAC the on the Humanoid task. To test the statistical significance of our result, we re-ran both SAC and OAC in a setting where 4 training steps per iteration are used. By exploiting the memory buffer more fully, the 4-step versions show the benefit of improved exploration

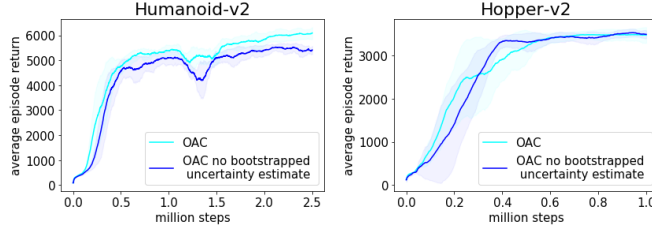

Figure 4: Impact of the bootstrapped uncertainty estimate on the performance of OAC.

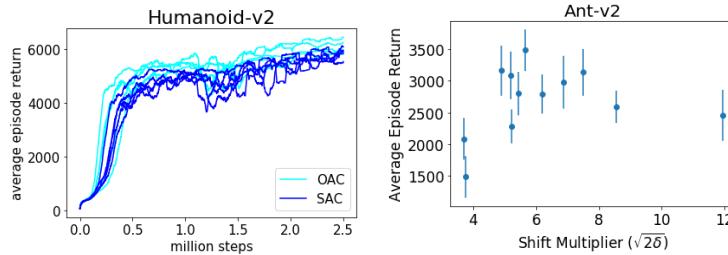

Figure 5: Left figure: individual runs of OAC vs SAC. Right figure: sensitivity to the KL constraint $\delta$. Error bars indicate $90\%$ confidence interval.

more clearly. The results are shown in the lower-right plot in Figure 3. At the end of training, the $90\%$ confidence interval[6] for the performance of OAC was $5033 \pm 147$ while the performance of SAC was $4586 \pm 117$. We stress that we did not tune hyper-parameters on the Humanoid environment. Overall, the fact that we are able to improve on Soft-Actor-Critic, which is currently the most sample-efficient model-free RL algorithm for continuous tasks shows that optimism can be leveraged to benefit sample efficiency. We provide an explicit plot of sample-efficiency in Appendix J.

**Usefulness of the Bootstrapped Uncertainty Estimate**   OAC uses an epistemic uncertainty estimate obtained using two bootstraps of the critic network. To investigate its benefit, we compare the performance of OAC to a modified version of the algorithm, which adjusts the exploration policy to maximize the approximate lower bound, replacing $\hat{Q}_{\text{UB}}$ with $\hat{Q}_{\text{LB}}$ in equation (9). While the modified algorithm does not use the uncertainty estimate, it still uses a shifted exploration policy, preferring actions that achieve higher state-action values. The results is shown in Figure 4 (we include more plots in Figure 8 in the Appendix). Using the bootstrapped uncertainty estimate improves performance on the most challenging Humanoid domain, while producing either a slight improvement or a no change in performance on others domains. Since the upper bound is computationally very cheap to obtain, we conclude that it is worthwhile to use it.

**Sensitivity to the KL constraint**   OAC relies on the hyperparameter $\delta$, which controls the maximum allowed KL divergence between the exploration policy and the target policy. In Figure 5, we evaluate how the term $\sqrt{2\delta}$ used in the the exploration policy (9) affects average performance of OAC trained for 1 million environment steps on the Ant-v2 domain. The results demonstrate that there is a broad range of settings for the hyperparameter $\delta$, which leads to good performance.

**Learning is Stable in Practice**   Since OAC explores with a shifted policy, it might at first be expected of having poorer learning stability relative to algorithms that use the target policy for exploration. While we have already shown above that the performance difference between OAC and SAC is statistically significant and not due to increased variance across runs, we now investigate stability further. In Figure 5 we compare individual learning runs across both algorithms. We conclude that OAC and SAC are similarly stable, avoiding the problems associated with stabilising deep off-policy RL [43, 47].

# 6 Conclusions

We present Optimistic Actor Critic (OAC), a model-free deep reinforcement learning algorithm which explores by maximizing an approximate confidence bound on the state-action value function. By addressing the inefficiencies of *pessimistic underexploration* and *directional uninformedness*, we are able to achieve state-of-the art sample efficiency in continuous control tasks. Our results suggest that the principle of optimism in the face of uncertainty can be used to improve the sample efficiency of policy gradient algorithms in a way which carries almost no additional computational overhead.

## Footnotes

[3]Policy improvement results can still be obtained with the entropy term present, in a certain idealized setting [24].

[4]In practice, the per-iteration wall clock time it takes to run OAC is the same as SAC.

[5]Because of this high variance, we measured a lower mean performance of SAC in Figure 3 than previously reported. We provide details in Appendix E.

[6]Due to computational constraints, we used a slightly different target update rate for OAC. We describe the details in Appendix D.

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
