[Supplementary Material · OAC-supplement-new-2-cr.pdf]

# Better Exploration with Optimistic Actor-Critic

## Appendices

## A   Proof of Proposition 1

**Proposition 1.** *The exploration policy resulting from* (6) *has the form* $\pi_E = \mathcal{N}(\mu_E, \Sigma_E)$, *where*

$$\mu_E = \mu_T + \frac{\sqrt{2\delta}}{\left\|\left[\nabla_a \hat{Q}_{UB}(s,a)\right]_{a=\mu_T}\right\|_{\Sigma_T}} \Sigma_T \left[\nabla_a \hat{Q}_{UB}(s,a)\right]_{a=\mu_T} \quad and \quad \Sigma_E = \Sigma_T. \qquad (10)$$

*Proof.* Consider the formula for the KL distance between two Gaussian distributions.

$$\text{KL}(\mathcal{N}(\mu,\Sigma), \mathcal{N}(\mu_T, \Sigma_T)) = \tfrac{1}{2}\left(\text{tr}(\Sigma_T^{-1}\Sigma - I) + \log\tfrac{\det\Sigma_T}{\det\Sigma} + (\mu-\mu_T)^\top \Sigma_T(\mu-\mu_T)\right). \quad (11)$$

Consider the minimization problem

$$\min_\Sigma \text{KL}(\mathcal{N}(\mu,\Sigma), \mathcal{N}(\mu_T, \Sigma_T)) = \min_\Sigma \tfrac{1}{2}\left(\text{tr}(\Sigma_T^{-1}\Sigma - I) + \log\tfrac{\det\Sigma_T}{\det\Sigma}\right). \qquad (12)$$

The equality in (12) follows by eliminating a term in KL that does not depend on $\Sigma$. From the fact that KL divergence is non-negative and because the term under the minimum becomes zero when setting $\Sigma = \Sigma_E$, we obtain:

$$\Sigma_E = \min_\Sigma \tfrac{1}{2}\left(\text{tr}(\Sigma_T^{-1}\Sigma - I) + \log\tfrac{\det\Sigma_T}{\det\Sigma}\right) = \Sigma_T.$$

Next, plugging $\Sigma_E = \Sigma_T$ into (6), we obtain the simpler optimization problem

$$\underset{\substack{\mu: \\ \text{KL}(\mathcal{N}(\mu,\Sigma_T),\mathcal{N}(\mu_T,\Sigma_T))\leq\delta}}{\arg\max} \bar{Q}_{\text{UB}}(s,\mu). \qquad (13)$$

Using the formula for the KL divergence between Gaussian policies with the same covariance, $\text{KL}(\mathcal{N}(\mu,\Sigma_T), \mathcal{N}(\mu_T, \Sigma_T)) = \tfrac{1}{2}(\mu-\mu_T)^\top \Sigma_T^{-1}(\mu-\mu_T)$, we can write (13) as

$$\text{maximize } \bar{Q}_{\text{UB}}(s,\mu) \text{ subject to } \frac{1}{2}(\mu-\mu_T)^\top \Sigma_T^{-1}(\mu-\mu_T) \leq \delta. \qquad (14)$$

To solve the problem (14), we introduce the Lagrangian:

$$L = \bar{Q}_{\text{UB}}(s,\mu) - \lambda(\frac{1}{2}(\mu-\mu_T)^\top \Sigma_T^{-1}(\mu-\mu_T) - \delta).$$

Differentiating the Lagrangian, we get:

$$\nabla_\mu L = \left[\nabla_a \bar{Q}_{\text{UB}}(s,a)\right]_{a=\mu} - \lambda\Sigma_T^{-1}(\mu-\mu_T)$$

Setting $\nabla_\mu L$ to zero, we obtain:

$$\mu = \frac{1}{\lambda}\Sigma_T \left[\nabla_a \bar{Q}_{\text{UB}}(s,a)\right]_{a=\mu} + \mu_T. \qquad (15)$$

By enforcing the KKT conditions, we obtain $\lambda > 0$ and

$$(\mu-\mu_T)^\top \Sigma_T^{-1}(\mu-\mu_T) = 2\delta. \qquad (16)$$

By plugging (15) into (16), we obtain

$$\lambda = \sqrt{\frac{\left[\nabla_a \bar{Q}_{\text{UB}}(s,a)\right]_{a=\mu_T}^\top \Sigma_T \left[\nabla_a \bar{Q}_{\text{UB}}(s,a)\right]_{a=\mu_T}}{2\delta}} = \frac{\left\|\left[\nabla_a \bar{Q}_{\text{UB}}(s,a)\right]_{a=\mu_T}\right\|_{\Sigma_T}}{\sqrt{2\delta}}. \qquad (17)$$

Plugging again into (15), we obtain the solution as

$$\mu_E = \mu_T + \frac{\sqrt{2\delta}}{\left\|\left[\nabla_a \bar{Q}_{\text{UB}}(s,a)\right]_{a=\mu_T}\right\|_{\Sigma_T}} \Sigma_T \left[\nabla_a \bar{Q}_{\text{UB}}(s,a)\right]_{a=\mu} = \qquad (18)$$

$$= \mu_T + \frac{\sqrt{2\delta}}{\left\|\left[\nabla_a \hat{Q}_{\text{UB}}(s,a)\right]_{a=\mu_T}\right\|_{\Sigma_T}} \Sigma_T \left[\nabla_a \hat{Q}_{\text{UB}}(s,a)\right]_{a=\mu}. \qquad (19)$$

Here, the last equality follows from (8). □

## B  Deterministic version of OAC

In response to feedback following the initial version of OAC, we developed a variant that explores using a deterministic policy. While using a deterministic policy for exploration may appear surprising, deterministic OAC works because taking an action that maximises an upper bound of the critic is often a better choice than taking the action that maximises the mean estimate. Formally, deterministic OAC explores with a policy that solves the optimization problem

$$\mu_E = \underset{\substack{\mu: \\ \mathrm{W}(\delta(\mu),\delta(\mu_T))\leq\delta}}{\arg\max} \bar{Q}_{\mathrm{UB}}(s,\mu). \tag{20}$$

Here, we denote a deterministic probability distribution (Dirac delta) centred at $a$ with $\delta(a)$. We used the Wasserstein divergence (denoted with W) because the KL metric becomes singular for deterministic policies. Below, we derive a result that provides us with an explicit form of the exploration policy for deterministic OAC, analogous to Proposition 1.

**Proposition 2.** *The exploration policy resulting from* (20) *has the form* $\pi_E = \delta(\mu_E)$, *where*

$$\mu_E = \mu_T + \frac{\sqrt{2\delta}}{\left\|\left[\nabla_a\hat{Q}_{UB}(s,a)\right]_{a=\mu_T}\right\|}\left[\nabla_a\hat{Q}_{UB}(s,a)\right]_{a=\mu_T}. \tag{21}$$

*Proof.* Consider the formula for the Wasserstein divergence between two Dirac-delta distributions.

$$\mathrm{W}(\delta(\mu),\delta(\mu_T)) = \tfrac{1}{2}\left\|\mu - \mu_T\right\|^2. \tag{22}$$

Plugging this into 20 leads to the optimization problem

$$\text{maximise } \bar{Q}_{\mathrm{UB}}(s,\mu) \text{ subject to } \frac{1}{2}(\mu - \mu_T)^\top(\mu - \mu_T) \leq \delta. \tag{23}$$

The rest of the proof follows analogously to the proof of Poposition 1, by observing that equation (23) is a special case of (14). □

The deterministic version of OAC is identical to regular OAC except for using the exploration policy from Proposition 2. An example experiment is shown in Figure 6.

Figure 6: Experiment showing the performance of a deterministic version of OAC.

## C  Population standard deviation of two values

For completeness, we include the computation of the population standard deviation of two values. Consider a two-element sample $\{\hat{Q}_{\mathrm{LB}}^1(s,a), \hat{Q}_{\mathrm{LB}}^2(s,a)\}$. Denote the sample mean by $\mu_Q(s,a) = \frac{1}{2}(\hat{Q}_{\mathrm{LB}}^1(s,a) + \hat{Q}_{\mathrm{LB}}^2(s,a))$. The population standard deviation takes the form

$$\sigma_Q(s,a) = \sqrt{\sum_{i\in\{1,2\}} \frac{1}{2}\left(\hat{Q}_{\text{LB}}^i(s,a) - \mu_Q(s,a)\right)^2} \qquad (24)$$

$$= \sqrt{\frac{1}{2}\left(\frac{1}{2}\hat{Q}_{\text{LB}}^1(s,a) - \frac{1}{2}\hat{Q}_{\text{LB}}^2(s,a)\right)^2 + \frac{1}{2}\left(\frac{1}{2}\hat{Q}_{\text{LB}}^1(s,a) - \frac{1}{2}\hat{Q}_{\text{LB}}^2(s,a)\right)^2} \qquad (25)$$

$$= \sqrt{\left(\frac{1}{2}\hat{Q}_{\text{LB}}^1(s,a) - \frac{1}{2}\hat{Q}_{\text{LB}}^2(s,a)\right)^2} \qquad (26)$$

$$= \frac{1}{2}\sqrt{\left(\hat{Q}_{\text{LB}}^1(s,a) - \hat{Q}_{\text{LB}}^2(s,a)\right)^2} \qquad (27)$$

$$= \frac{1}{2}\left|\hat{Q}_{\text{LB}}^1(s,a) - \hat{Q}_{\text{LB}}^2(s,a)\right| \qquad (28)$$

From this formula, we can obtain:

$$\mu_Q(s,a) - \sigma_Q(s,a) = \frac{1}{2}(\hat{Q}_{\text{LB}}^1(s,a) + \hat{Q}_{\text{LB}}^2(s,a)) - \frac{1}{2}\left|\hat{Q}_{\text{LB}}^1(s,a) - \hat{Q}_{\text{LB}}^2(s,a)\right| \qquad (29)$$

$$= \min(\hat{Q}_{\text{LB}}^1(s,a), \hat{Q}_{\text{LB}}^2(s,a)). \qquad (30)$$

This formula allows us to interpret the minimization used by TD3 [17] and SAC [24] as an approximate lower confidence bound. Similarly, we have:

$$\mu_Q(s,a) + \sigma_Q(s,a) = \max(\hat{Q}_{\text{LB}}^1(s,a), \hat{Q}_{\text{LB}}^2(s,a)). \qquad (31)$$

This indicates that using UCB with $\beta_{UB} = 1$ is equivalent to taking the maximum of the two critics.

## D   Experimental setup and hyper-parameters

Our learning curves show the total undiscounted return. Following the convention of SAC, we smoothen the curves, so the y-value at any point corresponds to the average across the last 100 data points.

Our implementation was based on `softlearning`, the official SAC implementation [25] [2]. All experiments for OAC and SAC were run within Docker containers on CPU-only $Standard\_D8s\_v3$ machine type on Azure Cloud.

We made the following modifications to the evaluation:

1. In addition to setting the initial seeds of the computational packages used (NumPy, Tensorflow [1]), we also fix the seeds of the environments to ensure reproducibility of results given initial seed values. In practice, the seed of the environment controls the initial state distribution.

2. Instead of randomly sampling training and evaluation seeds, we force the seeds to come from disjoint sets.

We list OAC-specific hyper-parameters in Table 1 and 2. The other parameters were set as provided in the implementation of the Soft Actor-Critic algorithms from the `softlearning` repository [2]. We list them in Appendix E for completeness.

The OAC hyper-parameters have been tuned on four Mujoco environments (Ant-v2, Walker2d-v2, HalfCheetah-v2, Hopper-v2), using Bayesian optimization with parameter ranges given in Table 1 and 2. We used the average performance over the first 250 thousand steps of the algorithm as the BO optimization metric. Dusring BO, we sampled 1 hyperparameter setting at a time and performed an experiment on a single seed. We did not tune hyperparameters on Humanoid-v2.

## E   Reproducing the Baselines

**Soft-Actor Critic**   SAC results are obtained by running official SAC implementation [25] from the `softlearning` repository.[2] Additionally, to ensure reproducibility, we made the same changes to

Table 1: OAC Hyperparameters when training with 1 training gradient per environment step

| Parameter | Value | Range used for tuning |
|---|---|---|
| shift multiplier $\sqrt{2\delta}$ | 6.86 | $[0, 12]$ |
| $\beta_{\text{UB}}$ | 4.66 | $[0, 7]$ |
| $\beta_{\text{LB}}$ | $-3.65$ | $[-7, -1]$ |

Table 2: OAC Hyperparameters when training with 4 gradient steps per environment step

| Parameter | Value | Range used for tuning |
|---|---|---|
| shift multiplier $\sqrt{2\delta}$ | 3.69 | $[0, 12]$ |
| $\beta_{\text{UB}}$ | 4.36 | $[0, 7]$ |
| $\beta_{\text{LB}}$ | $-2.54$ | $[-7, -1]$ |
| target smoothing coefficient $(\tau)$ | 0.003 | $[0.001, 0.005]$ |

the evaluation setup that were discussed in Appendix D. The SAC results we obtained are similar to previously reported results [25], except on HalfCheetah-v2. The performance of SAC on HalfCheetah-v2 in Figure 3 does not match the performance reported in [25]. This is because performance on this environment has high variance. We have also confirmed this observation with the SAC authors.

**TD3 and DDPG**  The official implementation of TD3 (`https://github.com/sfujim/TD3`) was used to generate the baselines in Figure 3. Similarly, the tuned DDPG implementation we use is the 'OurDDPG' baseline used in [17].

# F   Visualisations of the critic for continuous control domains

Figure 7: Visualization of the critic lower bound (red curve) and the action distribution for a given state. The policy mean $\mu$ is denoted by the light blue dot. Each of the 6 plot shows the values of the critic for a different state seen by the SAC agent training on Ant-v2.

Table 3: SAC Hyperparameters

| Parameter | Value |
|---|---|
| optimizer | Adam [30] |
| learning rate | $3 \cdot 10^{-4}$ |
| discount ($\gamma$) | 0.99 |
| replay buffer size | $10^6$ |
| number of hidden layers (all networks) | 2 |
| number of hidden units per layer | 256 |
| number of samples per minibatch | 256 |
| nonlinearity | ReLU |
| target smoothing coefficient ($\tau$) | 0.005 |
| target update interval | 1 |
| gradient steps | 1 |

Table 4: TD3 Hyperparameters (From [17])

| Parameter | Value |
|---|---|
| critic learning rate | 0.0003 |
| actor learning rate | 0.0003 |
| optimizer | Adam [30] |
| target smoothing coefficient ($\tau$) | 0.005 |
| target update interval | 1 |
| policy frequency | 2 |
| policy noise | 0.2 |
| noise clipping threshold | 0.5 |
| number of samples per minibatch | 100 |
| gradient steps | 1 |
| discount ($\gamma$) | 0.99 |
| replay buffer size | $10^6$ |
| number of hidden layers (all networks) | 1 |
| number of units per hidden layer | 256 |
| nonlinearity | ReLU |
| exploration policy | $\mathcal{N}(0, 0.1)$ |

Figure 7 shows the visualization of the critic lower bound and the policy during training. The 6 states used to plot were sampled randomly and come from the different training runs of Ant-v2. The second figure in the first row demonstrates the phenomenon of pessimistic underexploration in practice. The plots were obtained by intersecting a ray cast from the policy mean $\mu$ in a random direction with the hypercube defining the valid action range.

## G  Ablations

Figure 8 shows plots demonstrating the difference in the performance of OAC with and without the bootstrapped uncertainty estimate for all 5 domains.

## H  Policy gradient algorithms are greedy

In this section, we recall existing analytic results [13, 14] to provide more background about how policy gradient methods behave asymptotically. Following [14], we show the behavior of policy gradients on a simple task where the critic is a quadratic function, given by $\hat{Q} = -a^2$. In order to abstract away the effects of simulation noise, we use Expected Policy Gradients, a variance-free policy gradient algorithm where the policy at each step can be obtained analytically. In Figure 9, we

Table 5: DDPG Hyperparameters (From [17])

| Parameter | Value |
|---|---|
| critic learning rate | 0.001 |
| actor learning rate | 0.001 |
| optimizer | Adam [30] |
| target smoothing coefficient ($\tau$) | 0.005 |
| target update interval | 1 |
| number of samples per minibatch | 100 |
| gradient steps | 1 |
| discount ($\gamma$) | 0.99 |
| replay buffer size | $10^6$ |
| number of hidden layers (all networks) | 2 |
| number of units per hidden layer (from first to last) | 400, 300 |
| nonlinearity | ReLU |
| exploration policy | $\mathcal{N}(0, 0.1)$ |
| number of training runs | 5 |

Figure 8: Ablation studies for OAC on the benefit of the critic's approximate upper bound.

can see the policy variance as a function of optimization time-step. The fact that $\sigma^2$ decays to zero is consistent with the intuition that the function $-a^2$ has one optimum $a = 0$. The optimal policy simply puts an increasing amount of probability mass in that optimum. We empirically demonstrate in the middle plot in the upper row of Figure 7 that a similar phenomenon is happening with soft actor-critic. Since policy gradients are fundamentally optimization algorithms that maximize the critic, the policy covariance still becomes very small in the optima of the critic, despite entropy regularization and noisy sampling of actions used by SAC. While maximising the critic is actually a good thing if the critic is accurate, it leads to pessimistic under-exploration when we only have a loose lower bound. OAC addresses this problem by using a modified exploration policy.

## I   Upper bound encourages exploration

We give more intuitions about how using a lower bound of the critic inhibits exploration, while using an upper bound encourages it. Figure 10a shows the true critic, a lower bound and an upper bound for a one-dimensional continuous bandit problem. If we use the lower bound $\hat{Q}_{\text{LB}}$ for updating the policy, the standard deviation of the policy becomes small as described above, since we have a local maximum. On the other hand, using the upper bound $\hat{Q}_{\text{UB}}$, we have a moderate standard deviation and a strong policy gradient for the mean, meaning the policy doesn't get stuck. At first, one may think that the example in Figure 10a is cherry-picked since the upper bound may also have a spurious

Figure 9: Policy variance as a function of time, when applying policy gradients to a continuous bandit problem.

(a) Lower bound worse than upper bound for exploration.

(b) Smooth upper bound tight in a region near $\mu$ (here: to the right) cannot have a spurious local maximum at $\mu$.

Figure 10: Properties of the critic upper and lower bound for exploration.

local maximum. However, Figure 10b demonstrates that it cannot be the case. A smooth function cannot simultaneously track the true critic in some region close to $\mu$, have a spurious maximum at $\mu$ and be an upper bound.

## J   Plot of sample efficiency

Figure 11: Explicit Plot of Sample Efficiency

Figure 11 shows the number of steps required to achieve a given level of performance.

## K   Additional ablations

Figure 12a shows the shift, i.e. the norm of the difference between the mean of the target policy $\pi_T$ and the exploration policy $\pi_E$. The reported value is averaged across a single minibatch.

Figure 12b shows learning curves for four different values of $\beta_{UB}$, including the sweet-spot value of 4.36 that was used for the final result.

(a) Amount of policy shift at a given timestep.

(b) Effect of the optimism parameter $\beta_{UB}$.

Figure 12: Additional ablations.

## Footnotes

[2]Commit `1f6147c83b82b376ceed` , https://github.com/rail-berkeley/softlearning