[Reviews · NeurIPS 2019]

Reviewer 1



This work tackles the problem of sample efficiency through improved exploration. Off-policy Actor-Critic methods use the estimate of the critic to improve the actor. As they ignore the uncertainty in the estimation of the Q values, this may lead to sub-optimal behavior. This work suggests to focus exploration on actions with higher uncertainty (i.e., those which the critic is less confident about). The authors suggest of Optimistic Actor Critic approach, which is an extension to the Soft Actor Critic and evaluate their approach across various MuJoCo domains. This approach seems like a promising direction, the paper is well written and the motivation is clear, however I have several issues/questions: 1) Can this approach be applied to Actor-Critic schemes in discrete action spaces? The exploration scheme wouldn't end up as a shifted Gaussian, as in the continuous case, but rather a probabilistic distribution over the simplex which is bounded by distance \delta from the original policy. 2) Extension to non-convex action-value functions. This approach focuses on Gaussian policies, however, as the action-value function may be non-convex [1] considering Gaussian policies may result in convergence to local sub-optimal extremum. Can this approach be extended to more general policy distributions, such as a mixture of Gaussians (exists in the SAC source code and in a previous revision of their work)? 3) This work builds on-top of SAC, which is a maximum entropy approach. Since this is a general approach that can be applied to any off-policy Actor-Critic scheme, you should also show experiments with OAC applied to DDPG/TD3. 4) Evaluation: The results are nice, but aren't groundbreaking. The main improvement lies within the Hopper domain and the Humanoid 4-step version (in the rest it is unclear due to the variance in OAC and SAC). 5) Results for TD3 seem very low. Based on my experience (with TD3 in the v2 domains), it attains ~10k average on HalfCheetah, 4k on Ant and 2.5k on Hopper - with the official implementation (average over 10 seeds). [1] Actor-Expert: A Framework for using Q-learning in Continuous Action Spaces Small issues: 1) I wouldn't exactly call what TD3 do a lower bound. As they show, the basic approach results in over-estimation due to the self-bootstrap and their minimum approach simply overcomes this issue (similar to Double Q learning). This is different than an LCB which takes the estimated mean minus an uncertainty estimate. 2) line 162 - you refer to Eq. (12) which is defined in the appendix. Should refer to (5) instead. 3) At each step, computing the exploration policy requires calculating the gradient. It's OK that an approach is more computationally demanding, but it should be stated clearly. Note to the authors: I think this work identifies a real problem and the approach looks promising. However, I believe it there is additional work to do in order for this to become a full conference paper and this paper should definitely be accepted in the future. ------------- POST FEEDBACK --------------- The authors response satisfied most of my "complaints". Thank you for taking the time to provide full answers and additional experiments. I have updated my score accordingly. I urge the authors to continue working on this direction, expanding to discrete action spaces and other areas of interest which may benefit greatly from this approach and those similar to it.

Reviewer 2



Originality: off-course, exploration and actor-critic architectures are not new in reinforcement learning, but here the authors suggest to start from an original point of view (which is the combination of two main drawbacks or RL approaches (i) pessimistic under-exploration and (ii) directionally uninformedness). Quality: intuitions and ideas are both well developed and illustrated. The proposed algorithm is interesting as well, and definitely reaches good performances. However, experimental results show that, despite improvements regarding other methods, millions of environmental interactions are still required. So, even if the proposed approach is able to reach state of the art performances, the main problem, which is stated by the author in the first paragraph of the introduction, still remains (I quote) : "millions of environment interactions are needed to obtain a reasonably performant policy for control problems with moderate complexity". Of course, this remains one of the biggest challenges of (deep) RL. The paper is well positioned regarding the literature. Clarity: The paper is very well written and organized. Intuitions and ideas are well presented and illustrated. Significance: the topic is indeed interesting, and I think other researchers in the field will definitely be interested in the results of the paper. *** Post feedback *** Thanks to the authors for the feedback and additional explanations. I have updated the confidence score accordingly.

Reviewer 3



The paper proposes an original idea, which is brining the upper bound confidence estimate, typically used in bandits algorithms, to improve exploration in Actor-Critic methods. The claim that 33 "If the lower bound has a spurious maximum, the covariance of the policy will decrease" is not sustained or explained. The authors mentioned the problem with directional uninformedness and how SAC and TD3 sample actions in opposite directions from the mean with equal probability, but their proposal (5) is still a gaussian distribution, which just a different mean and variance, so the samples would be still have opposite directions from a different mean. Please clarify. The impact of the level of optimism is not analyzed in the experiments. In proposition 7, equation (7) it is not clear why the covariance matrices are the same. In Section 4.1, why not use the max{Q^1, Q^2} as the upper confidence bound of Q_{UB} instead of the mean + std? The claim that OAC improves sample complexity doesn't seem to follow from the experiments or figure 3, can you do a figure comparing steps to reach certain performance, as a way to show that? Minor comments: - In line 73 do you mean "Actor-Critic" instead of "Policy gradient"? - 80 \hat{Q}^1_{LB} -> \hat{Q}_{LB} - 162 There is not formula (12) - In Figure 3, add Humanoid-v2 1 training steps graph to the 4 training steps to make the comparison easier. - The paper uses a weird format for references that makes it harder to navigate them. ------------- POST FEEDBACK --------------- Thanks for to the authors for the explanations and for adding extra ablation studies and measuring the effect of the directionality. I have updated my overall score to reflect it.

[Author Response · NeurIPS 2019]

# **Better Exploration with Optimistic Actor-Critic**: Author Response

**All reviewers** Thanks for the feedback. As requested, we provide a plot measuring the sample efficiency gain (1) and additional ablations (Fig. 2 and 3). Also, OAC now supports deterministic policies as suggested by reviewer 1. While deterministic policies for exploration may appear surprising, deterministic OAC works because taking an action that maximises an upper bound of $Q^\pi$ is often a better choice than taking the action that maximises the mean estimate of $Q^\pi$. Results are in Fig. 4. Shaded bars denote one standard deviation (runs differ due to random initial state).

**Reviewer 1** Thanks for the careful review. You are making several relevant points. You suggest extending OAC to support discrete, multi-modal and deterministic policies. We followed your third suggestion. We extended the scope of Proposition 1 slightly to include the Wasserstein distance, deriving an OAC variant that works with deterministic policies. We report the experimental results in Figure 4, where deterministic OAC beats deterministic SAC. You also suggested (points 1 and 2 in *detailed comments*) extensions to discrete and multi-modal policies. On discrete policies, the alternative to Proposition 1 would, as you say, no longer shift the policy mean but instead constrain policy change over the probability simplex. On multimodal-policies, extending OAC to Gaussian mixtures akin to Actor-Expert (Lim, 2018) would imply taking the KL divergence between mixtures. We agree that these extensions are interesting, but they would be hard to pack into a single submission. We will discuss them in the future work section and also relate our algorithm to Actor-Expert (Lim, 2018). In point 3, you suggested making an optimistic variant of TD3 or DDPG. We agree this would be informative, but we had a limited computation budget and chose SAC because of its performance on Humanoid. On your point 4, as you say, our results may not be groundbreaking but the difference is statistically significant and a step forward (Fig. 1). Also, thanks for flagging the TD3 results. There was a problem in our setup and the results are now 5K on Ant after 2.5M steps. Concerning the *small issues* part of the review, we will clarify description of TD3, fix the misnumbered equation and discuss the cost of computing the additional gradient (it is very small in practice).

**Reviewer 2** We appreciate the kind words. You are right when you say that OAC still needs many environmental interactions. However, using OAC vs SAC does make a meaningful difference. On Humanoid, OAC obtains a policy of same quality in 0.52M steps vs 1M for SAC (Figure 1). We agree that improving sample efficiency remains a challenge and hope that OAC paves the way for even better methods.

**Reviewer 3** Thanks for the feedback. In *detailed comments*, you ask why a spurious maximum of the lower bound leads to a policy with small covariance. Intuitively, this is because the actor finds a probability distribution that greedily maximises the critic lower bound. But a distribution that maximises a function is a point mass at the maximum of that function. Formally, as actor iteration progresses, the covariance $\Sigma$ can be modelled as $e^{Ht}$, where $H$ is the second order term in the Taylor expansion of the critic around the policy mean and $t$ is the iteration count. Near a maximum, $H$ is negative definite and we have $\Sigma \propto e^{Ht} \to 0$ as $t \to \infty$. We will include an extension of this argument in the paper. Your second point concerns how our Gaussian exploration policy avoids directional uninformedness. This is best seen in the figure on page 5 of the paper. While the exploration policy $\pi_E$ is symmetric around its own mean, it is not symmetric around the mean of the target policy $\pi_T$. We will make this clearer. Also, you requested a measurement of the directionality. We provide it in Fig. 3, which tracks the absolute magnitude in the difference between the mean of the exploration policy and target policy. We also preformed an ablation for optimism, shown in Fig. 2. The figure shows a sweet spot (the optimism value $\beta_{\mathrm{UB}} = 4.36$ we used in the submission). About proposition 1, we will motivate $\Sigma_E = \Sigma_T$ more clearly. We will also expand the justification for this near line 450 of Appendix A. Also, you propose using $\max(\hat{Q}^1_{\mathrm{LB}}, \hat{Q}^2_{\mathrm{LB}})$ as the UCB. We in fact already do, i.e. $\max(\hat{Q}^1_{\mathrm{LB}}, \hat{Q}^2_{\mathrm{LB}}) = \mu_Q + \sigma_Q$, using notations from lines 154-158 and Appendix B. We will make this clearer. Also, as requested, we provide a plot measuring the number of steps to reach a given performance (Fig. 1). On your minor comments, we meant actor-critic in line 73. We will fix this as well as the typo, the misnumbered equation and the format of references.

Fig. 1: Sample efficiency

Fig. 2: Optimism ablation ($\beta_{\mathrm{UB}}$)

Fig. 3: Magnitude of shift.

Fig. 4: Deterministic OAC

## **References**

S. Lim, J. Ajin L. Le, Y. Pan, M. White *Actor-Expert: A Framework for using Action-Value Methods in Continuous Action Spaces*


[Meta-Review · NeurIPS 2019]

The paper addresses exploration in actor critic methods where the authors identify 2 main problems: pessimistic under-exploration and Directional uninformedness. The authors propose to use UCB upper and lower bounds based on the uncertainty of the value function. All reviewers appreciated the intuitive idea and the exhaustive evaluation of the approach. The results were also considered to be very promising and the authors provided additional ablation studies with their rebuttal. There was a consensus of all reviewers that the paper is a valueable contribution to the field of reinforcement learning.